# Monitoring the Coagulation Profile of COVID-19 Patients Using Standard and ClotPro^®^ Hemostasis Tests

**DOI:** 10.3390/medicina59071202

**Published:** 2023-06-26

**Authors:** Dragan Milić, Milan Lazarević, Natalija Vuković, Aleksandar Kamenov, Velimir Perić, Mlađan Golubović, Marija Stošić, Dimitrije Spasić, Vladimir Stojiljković, Dragana Stokanović

**Affiliations:** 1Medical School of Nis, University of Nis, 18000 Nis, Serbia; drdaganmilic@gmail.com (D.M.); kamenovcs@gmail.com (A.K.); velperic@gmail.com (V.P.); mladjangolubovic@gmail.com (M.G.); marija91nis@gmail.com (M.S.); serbvlada@gmail.com (V.S.); dragana.stokanovic@medfak.ni.ac.rs (D.S.); 2Clinic of Cardiovascular Surgery, University Clinical Center Nis, 18000 Nis, Serbia; dspasic@lec.edu; 3Clinic for Anesthesiology and Intensive Therapy, University Clinical Center Nis, 18000 Nis, Serbia; massha.vukovic@gmail.com

**Keywords:** COVID-19, coagulation, viscoelastic tests, ClotPro^®^, thrombosis, mortality

## Abstract

*Background and Objectives*: Coagulation disorders during COVID-19 infection are associated with a poorer prognosis and higher disease severity because thrombosis and inflammation are two processes that interfere with each other. A very important issue for clinicians is timely and adequate hemostasis and inflammation monitoring to prevent and treat potentially lethal consequences. The aim of this study was to identify specific hemostatic parameters that are associated with a higher risk of intrahospital mortality. *Materials and Methods*: This study was approved by the Ethics Committee of the Clinical Center Nis in Serbia. One hundred and forty-two patients presented with COVID-19 ARDS and were admitted to the ICU in the Clinic for Anesthesiology at the Clinical Center Nis from 14 April 2020 to 25 May 2020. Upon admission, blood was collected for biochemical and coagulation testing. The data obtained were analyzed using the Statistical Package for Social Sciences (SPSS v. 25, Chicago, IL, USA). *Results*: Among all the parameters assessed, older age; increased levels of fibrinogen, INR, D-dimer, and presepsin; and higher results in the platelet aggregation tests (aggregation induced by adenosine diphosphate based on the ADP test (AU/min), aggregation induced by arachidonic acid based on the ASPI test (AU/min), and aggregation induced by thrombin based on the TRAP test (AU/min)) and some assays of the viscoelastic test (clot amplitude after 5 min in the extrinsic coagulation pathway based on the A5 EX-test (mm), clot amplitude after 10 min in the extrinsic coagulation pathway based on the A10 EX-test (mm), clot amplitude after 5 min regarding functional fibrinogen based on the A5 FIB-test (mm), clot amplitude after 10 min regarding functional fibrinogen based on the A10 FIB-test (mm), and maximum clot firmness based on the MCF FIB-test (mm)); and lower values of viscoelastic clotting time in the extrinsic coagulation pathway based on the CT EX-test (s) were significantly correlated with mortality. In the multivariate analysis, D-dimer levels above 860 ng/mL, higher TRAP test value bins, and values above the normal reference range of the A10 FIB test were found to be independent predictors of mortality. *Conclusions*: Sophisticated hemostasis parameters can contribute to early risk assessment, which has initially been performed only on the basis of patients’ clinical status. Hypercoagulability is the main coagulation disorder in COVID-19 infection.

## 1. Introduction

The COVID-19 pandemic has been a challenge for healthcare systems around the world and is associated with the development of acute respiratory distress syndrome (ARDS), the need for admission to an intensive care unit (ICU), and a higher risk of death. Many different symptoms are present, but the most important are severe lung dysfunction, a need for ventilation, shock, and multiple organ failure [1].

Coagulation disorders during COVID-19 infection are associated with a poorer prognosis and higher disease severity because thrombosis and inflammation are two processes that interfere with each other [2]. Due to viral infection, pathogens initiate complex systemic inflammatory responses as part of innate immunity. Activation of the host immune system results in the activation of coagulation and thrombin generation in a process called immunothrombosis [3].

Inflammation is present in patients with SARS-CoV-2 infection, and levels of IL-6, C-reactive protein, procalcitonin, and fibrinogen are elevated [4]. Endothelial cell activation and damage result in disruption of the natural antithrombotic state [5]. This inflammation and activation of coagulation are the causes of elevated D-dimer levels, as increased levels have been associated with thromboembolism [6]. Some patients have systemic inflammatory response syndrome (SIRS) or cytokine storm, which may explain the more dramatic changes in coagulation tests, including significantly elevated D-dimer levels and changes in other hemostasis tests, especially as the disease progresses [7,8].

The receptor that the SARS-CoV-2 virus adheres to is the angiotensin-converting enzyme 2 receptor present on endothelial cells, and viral replication causes inflammatory cell infiltration, endothelial cell death, and microvascular thrombosis [9]. As a result, microcirculatory dysfunction contributes to the clinical symptoms in patients with COVID-19.

The aim of the research is to: (1) examine the ability of routine parameters of hemostasis and the Clot Pro test to predict mortality; (2) examine the association of the examined parameters; (3) find the cutoff values of parameters that show predictive potential; and (4) test the predictive ability of combinations of obtained parameters. We assessed intrahospital mortality and correlated hemostasis parameters with it. During this research, we used classic coagulometric tests of hemostasis, which included prothrombin time, activated partial thromboplastin time, fibrinogen concentration, d-dimer concentration, and anti-Xa values. On the other hand, point-of-care (POC) hemostasis tests were also used.

Critical illness is known to cause a procoagulable state due to immobilization, mechanical ventilation, and central venous access, but COVID-19 can cause a hypercoagulable state with mechanisms unique to the virus and cross-talk between thrombosis and inflammation. This is why the bedside POC hemostasis test is very important for the quick identification of coagulation disturbances and clinical treatment. We used various tests, such as the viscoelastic global hemostasis test (ClotPro^®^), while also testing for platelet function using impedance aggregometry (Multiplate analyzer). Viscoelastic testing can be beneficial in clinical practice. Extrinsic and intrinsic coagulation pathways, as well as hyperfibrinolysis, can be evaluated using clot formation time and clot firmness as markers (MCF) in ClotPro^®^. [10]. An additional test with tissue plasminogen activator (tPA) to initiate fibrinolysis might be a promising technique to detect hyperfibrinolysis [11], but we could not use it because it is not registered in our country.

Recent studies have shown a correlation between thromboembolic events and abnormal results in the global hemostasis test, ClotPro^®^, especially when combined with D-dimer values [12]. It remains unclear whether this could be helpful to predict or even prevent arterial or venous thromboembolic complications via the adjustment of anticoagulation therapy.

Impedance aggregometry is most important for accurate monitoring of platelet aggregation and is fundamental for guiding clinicians during anticoagulant and antiplatelet therapies because platelet hyperreactivity that follows a proinflammatory state, such as COVID-19 infection, can lead to higher complications and mortality [13,14].

Hypercoagulability is also present in cancer, autoimmune diseases, celiac disease (CD), and inflammatory bowel diseases. For example, in celiac disease, malabsorption caused by CD often leads to vitamin K deficiencies [15]. Vitamin K is a cofactor for the synthesis of protein C and protein S, and deficiency leads to uncontrolled coagulation-activated thrombosis [16]. Cancer-related hypercoagulability results from a lack of endogenous heparin to maintain the blood in its liquid form because of the degradation of endogenous heparin by tumor-secreted heparanase [17]. In cancer patients, thrombosis can also occur due to poor performance status, stasis, or drug-associated thrombosis. Patients with autoimmune disorders can develop antibodies against various phospholipids (APS), an auto-immune condition associated with thrombosis [18]. Heparin-induced thrombotic thrombocytopenia, immune thrombotic thrombocytopenic purpura, and vaccine-associated thrombotic thrombocytopenia are also disorders that can lead to hypercoagulability.

A very important issue for clinicians is timely and adequate hemostasis and inflammation monitoring to prevent and treat potentially lethal consequences. Given that a COVID-19 infection could have very serious consequences, including increased mortality of patients, often as a result of thromboembolic complications, we realized that routine hemostasis testing is insufficient to result in a better treatment outcome and that we needed fast and more modern descriptive tests that would give clinicians a better insight into potential risks. This was also the reason why we conducted this research, in the course of which we managed to help patients with additional medical measures.

## 2. Materials and Methods

This prospective study was conducted in accordance with the Declaration of Helsinki with research ethics board approval, and informed consent was obtained from all subjects involved in the study. One hundred forty-two patients presented with COVID-19 ARDS and were admitted to the ICU in the Clinic for Anesthesiology at the Clinical Center Nis from 14 April 2020 to 25 May 2020. Data from COVID-19 adult patients confirmed by RT-PCR were retrieved and analyzed. Prior to their admission to the ICU, all patients had COVID-19 symptoms for 5–7 days. The exclusion criteria were patients with pre-existing thrombotic or bleeding disorders, patients with pre-existing acquired coagulopathies, patients on chemotherapy or with active malignancy, and patients with renal disease, autoimmune diseases, or pregnancy. Patients developed some of the above-mentioned conditions (AKI—acute kidney injury) as a complication of a critical illness. Clinical variables monitored during the study were venous thromboembolism, pulmonary thromboembolism, and arterial thrombosis. All patients were critical (patients who were categorized as having severe to critical disease according to the WHO classification of COVID-19 disease and admitted to a critical care unit) [14]. All patients, aged from 36 to 84 years, including both females (38%) and males (62%), who were under tracheal intubation and mechanical ventilation, were enrolled in the study, and all received a prophylactic dose of low-molecular-weight heparin (LMWH) according to their body mass. Patients were sedated by continuous administration of propofol or dexmedetomidine, with monitoring of the depth of sedation by the bispectral index (BIS). Continuous administration of a non-depolarizing muscle relaxant was not routinely applied. All patients were ventilated in a semi-recumbent position using a closed suctioning system. In order to prevent pressure ulcers, all patients were turned every two hours. All patients received a daily dose of 40 to 80 milligrams of pantoprazole intravenously. Enteral nutrition was applied within 48 hours of ICU admission, except in the presence of clear contraindications for it. All patients had some type of corticosteroid therapy (dexamethasone, methylprednisolone, or hydrocortisone, which was most often given in septic shock until hemodynamic stabilization). Exposing patients to empiric antimicrobial therapy was performed for the shortest time until the arrival of sampled cultures and conditions for de-escalation. The average length of a stay in the ICU was 8 days. On admission to the ICU, venous blood was randomly collected for biochemical and coagulation testing.

Blood samples were taken from the antecubital vein and stored in serum vacutainer tubes without additives for c-reactive protein (CRP ng/mL), using the immunoturbidimetry method on a Beckman Coulter AU 680 analyzer (Beckman Coulter Inc., Brea, CA, USA).

Presepsin (PSEP) (pg/mL) levels were measured with 4 mL whole blood specimens using chemiluminescence enzyme immunoassay and Magtration^®^ technology on a PATHFAST Immunoanalyzer (Mitsubishi Chemical Europe GmbH, Düsseldorf, Germany).

For coagulation profile sample testing (D-dimer in ng/mL, prothrombin time (PT) in seconds, international normalized ratio (INR), activated partial thromboplastin time (aPTT) in seconds, fibrinogen concentration in g/L, and anti-Xa values), we used 4 mL whole blood citrated tubes, and the tests were performed using an ACL TOP 350 coagulometer (Instrumentation Laboratory, Bedford, MA, USA).

A viscoelastic test (Clot Pro^®^, Enicor, Germany) was performed using a rotary thromboelastometry device with 4 mL of whole blood in a test tube and sodium citrate as an anticoagulant, and the blood samples were also analyzed within 30 min of sampling. The outputs of the instrument consist of the following: (1) coagulation time (CT, which depends on the concentration and activity of coagulation factors in plasma, measured in seconds); (2) clot amplitude after 5 and 10 min (A5 and A10; clot also depends on platelet count/function and fibrinogen concentration, measured in mm); and (3) maximum clot firmness (MCF, whose value is determined based on the number and platelet function as well as fibrinogen concentration, measured in mm), using both the EX-test (the external coagulation pathway) and FIB-test (functional fibrinogen test).

For impedance aggregometry based on platelet function testing (Multiplate, Roch, Germany) we collected blood in 4 mL lithium heparinized tubes, and we used different platelet agonists in three separate tests to measure platelet aggregation: (1) aggregation with adenosine diphosphate in the ADP test (aggregation units per minute, AU/min), which is sensitive to ADP blocker therapy or Glanzman thrombastenia; (2) aggregation with arachidonic acid in the ASPI test (aggregation units per minute, AU/min), which sensitive to acetylsalicylic acid and NSAIL; and (3) aggregation with thrombin in the TRAP test (aggregation units per minute, AU/min), which represents natural platelet aggregation potential.

For whole blood count measurement, we used a Horiba ABX 200 (Horiba Medical, Palaiseau, France) counter, and blood was drawn in 4 mL tubes with ethylene di-amine tetra-acetic acid (EDTA).

### Statistical Analysis

The data obtained were analyzed using the Statistical Package for Social Sciences (SPSS v. 25, Chicago, IL, USA). According to the normality of distribution, continuous variables are presented as means with a standard deviation or as medians with an interquartile range. Categorical variables are presented as absolute and relative numbers. The differences between the two tested groups were tested using the parametric Student’s t-test, the non-parametric Mann-Whitney U test, and Fischer’s exact test. The correlation between continuous variables was assessed according to Pearson’s correlation coefficient. Univariate and multivariate binary logistic regressions were performed to determine statistically significant predictors of the dependent variables. We evaluated the discriminatory power of various laboratory parameters and determined the optimal cutoff values using receiver operating characteristic (ROC) curve analyses. The ROC curves for multiple variables were constructed based on the probabilities obtained via binary logistic regression modeling and compared with the DeLong test using MedCalc (v. 19.0; MedCalc Software Ltd., Ostend, Belgium). A *p*-value less than 0.05 was considered to indicate statistical significance.

## 3. Results

### 3.1. Inflammation and Coagulation Parameters and Mortality

Among all parameters assessed, older age; increased levels of fibrinogen, INR, D-dimer, and PSEP; higher results of platelet aggregation tests (ADP, ASPI, and TRAP); some assays of the viscoelastic test (ClotPro^®^) regarding clot firmness (A5 EX-test, A10 EX-test, A5 FIB, A10 FIB, and MCF FIB); and lower values of the viscoelastic CT EX-test were significantly correlated with higher mortality (Table 1).

When we analyzed all of these parameters from the perspective of their normal ranges, higher mortality was associated with a normal range of fibrinogen concentration rather than with values above this range (*p* < 0.05).

A total of 38 patients died in the ICU, and among them, 11 (28.95%) patients developed venous thromboembolism and pulmonary embolism, and 3 (7.9%) patients died after arterial thrombosis besides prophylaxis with LMWH. These patients did not receive antithrombotic therapy.

Values below the normal range were associated with survival in cases of ADP (*p* < 0.001), ASPI (*p* < 0.001), and TRAP tests (*p* < 0.001). On the contrary, death occurrence was more frequent in patients with D-Dimer (0 < 0.001), PSEP (*p* < 0.001), A10 FIB-test (*p* < 0.001), and MCF FIB-test values above the normal range (*p* < 0.01). Good clinical outcome, in terms of better survival, was associated with higher ranges (normal or above normal) of the CT EX-test (*p* < 0.05) but lower ranges (normal or below normal) of the A10 EX-test (*p* < 0.05).

Higher mortality was detected with extreme values of D-Dimer above 1000 ng/mL (*p* < 0.001), ADP test above 590 AU/min (*p* < 0.001), ASPI test above 800 AU/min (*p* < 0.001), TRAP test above 1500 AU/min (*p* < 0.001), and PSEP above 1000 pg/mL (*p* < 0.05).

### 3.2. Cutoff Values of Tests to Discriminate Patient Mortality

Using the ROC analysis, we identified the optimal cutoff values for a number of inflammation and coagulation parameters with the highest sensitivity and specificity in discriminating patients who died later (Figure 1).

Good discriminatory ability (AUC > 0.7) was shown for the following parameters: fibrinogen (≥9.14 g/L), INR (≥1.38), PSEP (≥335 ng/mL), A5 FIB-test (≥28 mm), and MCF FIB-test (≥36 mm). The cutoff values for the ADP test (≥591 AU/min) and ASPI test (≥728 AU/min) were excellent in discriminating patients with exitus. The best discriminators, with AUC > 0.9, were D-Dimer (≥860 ng/mL), TRAP (≥1180 AU/min), and A10 FIB-test (≥30 mm), which were all significantly better than the other parameters but without statistically significant differences among them.

After making combinations of two or three of the best discriminatory parameters, we found that the combination of D-Dimer and TRAP test was worse than other two-parameter combinations (*p* < 0.05), but the combinations of A10 FIB-test with the other two parameters were equally good for all the two- or three-parameter combinations tested (Table 2).

### 3.3. Correlation between Analyzed Parameters

We found various degrees of correlation between the analyzed variables (Table 3). There was a strong positive correlation among the ADP, ASPI, and TRAP tests (*p* < 0.001), as well as among the A5 EX-test, A10 EX-test, and MCF EX-test (*p* < 0.001), and between the A5 FIB-test and MCF FIB-test (*p* < 0.001). Strong negative correlation existed between the A10 EX-test (*p* < 0.001) and MCF EX-test (*p* < 0.001).

Presepsin concentration had a weak positive correlation with the A10 FIB-test (*p* < 0.001). Fibrinogen moderately correlated with the A5 FIB-test (*p* < 0.001) and MCF FIB-test (*p* < 0.001); additionally, its weak positive correlation was noted with the A5 EX-test (*p* < 0.001), A10 EX-test (*p* < 0.001), MCF EX-test (*p* < 0.001), and CT FIB-test (*p* < 0.001).

### 3.4. Predictors of Mortality

After performing the univariate binary logistic regression (Table 4) to find predictors of higher mortality, a number of parameters stood out. In cases where the same parameter was found to be significant in various forms (continuous variable, according to the normal range, or according to the previously found cutoff value), the form with the highest predictive value was chosen to be included in the multivariate model. The following variables’ cutoff values, as previously determined, had a higher predictive value than the absolute values or standard cutoffs: age, fibrinogen, INR, D-Dimer, ASPI test, PSEP, A5 EX-test, A10 EX-test, and A5 FIB-test.

Interestingly, the TRAP test and A10 FIB-test cutoff values, previously shown to have high discriminatory ability, were less valuable in the logistic regression modeling of mortality. Due to the total patient number and the high collinearity between some variables, the number of predictors in the multivariate model had to be reduced. The most fitted multivariate model (χ2 = 141.007, *p* < 0.001) explains 63.0–91.6% of the variance in death occurrence. All three variables included in the model were found to be independent predictors of mortality. D-Dimer above 860 ng/mL increased the risk of death by 24 times (*p* < 0.01). The TRAP test values were binned according to the normal value range, and with each higher value bin, the risk was 22 times higher (*p* < 0.01). A10 FIB-test values above the normal range resulted in a 290-fold greater risk of death (*p* < 0.05).

## 4. Discussion

Extensive activation of the coagulation cascade leads to disseminated intravascular coagulation and thrombosis, the inevitable progression of COVID-19, and mortality. In the background of these two extremes are hypercoagulability and aberrant fibrinolysis, where extreme values of PT, aPTT, platelet count, fibrinogen, and fibrin can be expected and are associated with COVID-19 mortality [19]. Our aim was to identify parameters that could be markers of the aforementioned pathophysiological mechanisms and determine the predictive cutoff values for these parameters that would enable sufficiently early detection and stratification of COVID-19 patients who were most at risk of death.

Endothelial injury and subsequent tissue factor genesis, as well as inhibited fibrinolysis due to changes in the concentrations of urokinase-type plasminogen activator and plasminogen activator inhibitor-1, are the main pathophysiological mechanisms of focal or disseminated intravascular coagulation [20]. Severe endothelial injury and subsequent vascular thrombosis and angiogenesis are the three principal morphological findings in COVID-19-related acute respiratory distress syndrome [21].

Vascular injury reflects extensive D-dimer elevations [22]. Therefore, this marker is recognized by the International Society of Thrombosis and Haemostasis (ISTH) as the most important among the data we receive from routine initial analysis in patient risk stratification [23].

The concordance (C) statistic with the value of the area under the ROC (receiver operating characteristic) curve (AUC) is a gold standard for outcome prediction for different predictive models. A generally accepted value for having excellent predictive ability for some diagnostic tests is 0.8 [24].

In our study, three variables resulted in extraordinary discriminatory capacity with an AUC ˃ 0.9, namely the calculated cutoff values of the D-dimer, the thrombin receptor-activating peptide (TRAP) test, and A10 in FIBTEM. These parameters were the basis for creating predictive models for estimating the risk of mortality in the most severe COVID-19 patients. Due to the hypofibrinolytic profile in thromboelastometry, there has been concern about the predictive ability of D-dimer [20]. The predictive ability of D-dimer increases over time. This fact should be kept in mind when analyzing the results since our research included patients with already developed ARDS [25].

In the literature, there are different values of D-dimer that are associated with poor patient outcomes. In our research, the cutoff value was 860 ng/mL. In their study, Liliana Baroiu et al. reported an average value of D-dimers of 841.22 ng/mL, with an SD of 1160.85 ng/mL and a median value of 473.73 ng/mL in the total group. The average value of D-dimers in the unfavorable evolution group (2057 ng/mL) was statistically significantly higher than the average D-dimers in the favorable evolution group (784 ng/mL; *p* < 0.0001) [26].

Despite the characteristic hypercoagulable profile, decreased CFT and increased MCF were thromboelastometrically confirmed, and there is no data on the correlation between thromboelastometric parameters and clinical outcomes [27]. In our study, the A10 FIB-test stood out as the best predictive marker of mortality outcome among the data obtained via thromboelastometry. A10 depends on platelet count and function and fibrinogen concentration. Fibrinogen is a basic substrate for hemostasis, and platelets are responsible for primary hemostasis; thus, clot amplitude after 10 min is very informative and is a significant test for evaluating the coagulation process. Not only does it correlate with the MCF value of the same test, but the A10 FIB-test has proven to be more useful in predicting different clinical outcomes. It belongs to the group of markers that depict the strength of clots and shows the strongest predictive potential compared to other FIB-test elements that are generally elevated in hypercoagulable states, such as during COVID-19 infection. The clinical advantage of this marker would be its rapid detection, both in relation to MCF and in relation to conventional laboratory tests. Given these advantages, the A10 FIB-test could be a useful parameter when admitting patients to an ICU [28,29].

Heubner L. et al. showed in their research that viscoelastic POC coagulation tests could help detect a hypercoagulation state and fibrinolysis in ARDS patients. They proved that patients with increased clot firmness (MCF) were more likely to present severe ARDS, which was also detectable in our results [30].

In contrast to classical coagulometric tests that reflect a specific pathway of coagulation, viscoelastic assays such as ClotPro^®^ provide a global assessment of changes in clot characteristics, from the initiation of the clot to stability and lysis. In ClotPro^®^, viscoelastic assays can be used to assess hypercoagulability. In patients with COVID-19, various groups have noted a shorter time for clot generation (CT), elevated fibrinogen concentration and function (A10), increased maximum clot firmness (MCF), and increased clot stability [31]. Many researchers performed viscoelastic tests on patients with COVID-19 in studies that enrolled only 20–50 patients. Although definitions and cutoff values vary, a consistent issue is the detection of hypercoagulability via increased fibrin clot strength (MCF), despite anticoagulant prophylaxis.

Johannes Herrmann et al. showed that thromboelastometry is a very useful tool for detecting hypercoagulation in COVID-19 patients, especially when monitoring the A10 parameter and MCF. From the first day, this basic parameter can predict thrombosis values and recovery after two weeks. Our results also indicated the significance of the A10 parameter in the FIB-test due to the enormous role of fibrinogen in promoting coagulation [32].

An elevated level of the TRAP test has great predictive potential for intrahospital mortality in COVID-19 patients. This result can be very useful in daily clinical practice, considering that the TRAP test represents baseline platelet aggregation and is independent of the influence of acetylsalicylic acid derivatives and P2Y12 inhibitors. We interpret high TRAP test levels as indicating an overactivated phenotype of platelets, which may be associated with a hypercoagulable state, disease progression, and mortality [33].

Although the mechanisms of platelet activation in COVID-19 remain unknown, existing research shows that inflammatory and procoagulant mediators in COVID-19 patients may contribute to platelet activation. Procoagulant factors, such as thrombin, may contribute to TF expression [34].

COVID-19 infection is characterized by impaired coagulation that can increase mortality. Platelets are fast responders to pathogen presence, and they contribute to thrombosis. The SARS-CoV-2 genome has been found in platelets from patients with COVID-19. Ellinor I. et al. in their research discuss platelet activation and immunothrombosis in patients with COVID-19, the effect of Spike on platelets, the activation of platelets by classical platelet activation triggers, as well as the contribution of platelets to complement activation [35].

Presepsin is a biomarker that has been studied in relation to sepsis, a potentially life-threatening condition that occurs when the body’s response to infection causes damage to its own tissues and organs. There is research evidence that suggests that presepsin levels may be elevated in individuals with severe COVID-19, particularly those who develop sepsis as a complication of the disease [36].

Bacteremic co-infection is a leading cause of ICU admission, mechanical ventilation, and mortality in individuals with COVID-19, and studies have shown that patients with severe COVID-19 who develop sepsis have a higher risk of death compared to those who do not. Due to the similar immunological and pathophysiological backgrounds of sepsis and COVID-19 and frequent bacterial co-infection, it is logical to test the predictive ability of presepsin. High values of presepsin may be a useful tool in predicting which individuals with COVID-19 are at higher risk of developing sepsis and potentially dying from the disease [37]. However, the cutoff value of presepsin that we obtained is slightly above the upper limit and is significantly lower compared to other studies.

The relationship between presepsin and COVID-19 mortality is still being studied, and more research is needed to fully understand the extent of this connection. Other factors, such as age, underlying health conditions, and access to medical care, may also play a role in determining an individual’s risk of mortality from COVID-19 [38].

All of the findings above have been very helpful for everyday clinical practice in the treatment of COVID-19 patients because we are able to detect hemostasis disturbances and react appropriately with proper anticoagulation and antithrombotic therapy to prevent and treat complications.

### Limitations of the Study

The interpretation of VET results in COVID-19 patients is still under discussion. The results obtained from different viscoelastic devices are not always interchangeable (due to different technologies). There are no clear cutoff values for the diagnosis of either clinically relevant hypercoagulability or hypofibrinolysis. We did not perform the tPA test, which is very important for detecting impaired hyperfibrinolysis, because it is not a registered test in our country. We also did not compare our results to those of a healthy control group.

## 5. Conclusions

Sophisticated hemostasis parameters can contribute to early risk assessment, which was initially performed only on the basis of patients’ clinical picture. Hypercoagulability is the main coagulation disorder in COVID-19 infection.

Conventional coagulation tests, such as PT (INR) and activated partial thromboplastin time (aPTT), present results for the initiation of the clotting process. However, these coagulation tests do not detect hypercoagulability, which is the most important issue in COVID-19 patients. Assessment of coagulation using conventional tests in COVID-19 patients is insufficient. D-dimer reflects an increased coagulation process, but it is non-specific. Few studies have identified D-dimer as a prognostic marker for clinical severity and mortality in COVID-19 patients. Thromboelastometry ClotPro^®^ is a point-of-care test that evaluates clot initiation, formation, stabilization, and lysis. Hypercoagulability occurs in COVID-19 patients, and its severity is identified based on clot strength in ClotPro^®^.

The natural platelet aggregation potential (TRAP) test is also a very fast and informative test for detecting platelet hyperreactivity and potential causes of arterial and venous thromboembolism.

In terms of the tested point-of-care parameters, D-dimer, the A10 FIB-test, and the TRAP test stand out, and their combinations are characterized by outstanding predictive potential for the detection of in-hospital mortality from COVID-19 infection. These parameters are easy to interpret and readily available, are elements of different hemostasis tests, and are markers of different effects on hypercoagulability.

Larger multicenter studies are needed in order to fully define the role of these tests in predicting patients’ procoagulant state during COVID-19 infection, but we are certainly working on good starting points that are increasingly applied in everyday clinical practice.

## Figures and Tables

**Figure 1 medicina-59-01202-f001:**
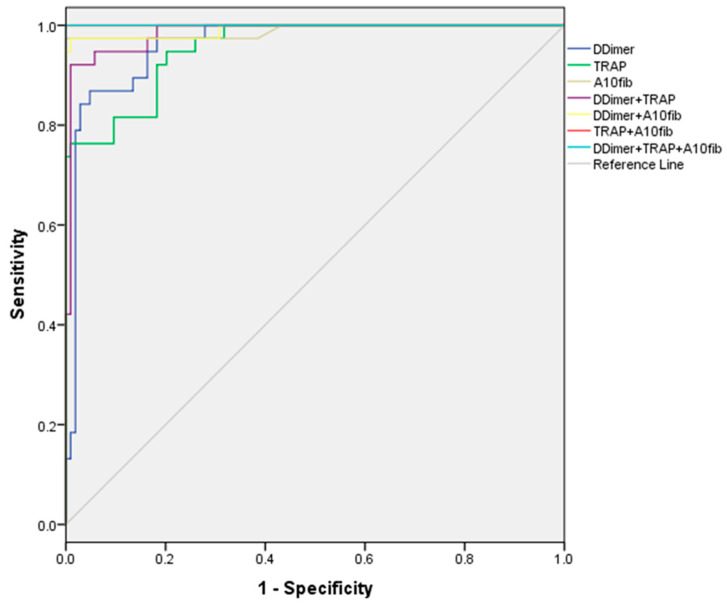
ROC analysis: sensitivity and specificity of inflammation and coagulation parameters.

**Table 1 medicina-59-01202-t001:** Inflammation and coagulation parameters according to mortality.

		Survival (N = 104)	Mortality (N = 38)	t * or Z ** or χ2 *** (*p*)
Age (years), mean ± SD		62.67 ± 12.10	66.71 ± 8.44	2.229 (0.028) *
Fibrinogen concentration (2–4 g/L), mean ± SD		7.62 ± 1.54	8.56 ± 2.29	2.344 (0.023) *
Fibrinogen concentration (>4 g/L), no. of patients (%)		103 (99.0%)	34 (89.5%)	4.944 (0.018) ***
Anti-Xa values (0.7–1), mean ± SD		0.38 ± 0.14	0.37 ± 0.15	0.323 (0.747) *
INR (0.85–1.25), mean ± SD		1.26 ± 0.16	1.48 ± 0.23	5.500 (˂0.0001) *
INR (>1), no. of patients (%)		102 (98.1%)	38 (100.0%)	0.003 (1.000) ***
D-Dimer (−230 ng/mL), mean ± SD		444.0 (407.0–737.5)	1285.0 (970.0–1542.0)	8.393 (˂0.0001) **
D-Dimer (>230 ng/mL), no. of patients (%)		89 (85.6%)	38 (100.0%)	4.697 (0.011) ***
D-Dimer (>1000 ng/mL), no. of patients (%)		2 (1.9%)	27 (71.1%)	77.640 (˂0.0001) ***
aPTT (29–34 s), mean ± SD		37.53 ± 8.92	36.38 ± 7.14	0.712 (0.478) *
aPTT (s), no. of patients (%)	<34	45 (43.3%)	20 (52.6%)	1.348 (0.510) ***
	34–38	14 (13.5%)	3 (7.9%)	
	>38	45 (43.3%)	15 (39.5%)	
ADP test (590–1130 AU/min), mean ± SD		365.14 ± 150.88	670.21 ± 224.77	7.753 (˂0.0001) *
ADP test (AU/min), no. of patients (%)	<406	61 (58.7%)	3 (7.9%)	37.248 (˂0.0001) ***
	406–992	43 (41.3%)	30 (78.9%)	
	>992	0 (0.0%)	5 (13.2%)	
ADP test (AU/min) (>590), no. of patients (%)		5 (4.8%)	24 (63.2%)	54.771 (˂0.0001) ***
ASPI test (790–1490 AU/min), mean ± SD		486.61 ± 229.59	843.84 ± 217.04	8.326 (˂0.0001) *
ASPI test (<790 AU/min), no. of patients (%)		103 (99.0%)	15 (39.5%)	66.129 (˂0.0001) ***
ASPI test (>800 AU/min), no. of patients (%)		0 (0.0%)	22 (57.9%)	66.896 (˂0.0001) ***
TRAP test (923–1509 AU/min), mean ± SD		548.25 ± 293.04	1375.90 ± 367.61	13.884 (˂0.0001) *
TRAP test (AU/min),	<923	94 (90.4%)	7 (18.4%)	78.997 (˂0.0001) ***
	923–1509	10 (9.6%)	14 (36.8%)	
	>1509	0 (0.0%)	17 (44.7%)	
TRAP test (>1500 AU/min), no. of patients (%)		0 (0.0%)	20 (52.6%)	59.435 (˂0.0001) ***
PSEP (˂337 pg/mL), mean		293.0 (239.0–395.5)	593.0 (446.2–743.8)	5.074 (˂0.0001) **
PSEP (>337 pg/mL), no. of patients (%)		27 (26.0%)	33 (86.8%)	39.818 (˂0.0001) ***
PSEP (>1000 pg/mL), no. of patients (%)		0 (0.0%)	3 (7.9%)	5.005 (0.018) ***
CT ex-test (s), mean ± SD		65.38 ± 16.38	57.58 ± 15.77	2.535 (0.012) *
CT EX-test (38–65 s), no. of patients (%)	<38	1 (1.0%)	3 (7.9%)	8.958 (0.011) ***
	38–65	47 (45.2%)	23 (60.5%)	
	>65	56 (53.8%)	12 (31.6%)	
A5 EX-test (39–58 mm), mean		53.0 (47.0–56.0)	56.5 (51.0–60.0)	2.628 (0.009) **
A5 EX-test (<39 mm), no. of patients (%)		7 (6.7%)	0 (0.0%)	1.446 (0.190) ***
A10 EX-test (47–64 mm), mean		59.5 (55.0–63.0)	63.5 (58.5–66.0)	3.174 (0.002) **
A10 EX-test (mm), no. of patients (%)	<38	10 (9.6%)	0 (0.0%)	6.415 (0.040) ***
	38–64	89 (85.6%)	33 (86.8%)	
	>64	5 (4.8%)	5 (13.2%)	
MCF EX-test (53–67 mm), mean		61.0 (58.0–64.0)	63.0 (60.0–67.2)	2.405 (0.016) **
MCF EX-test (mm), no. of patients (%)	<53	4 (3.8%)	0 (0.0%)	2.099 (0.350) ***
	53–68	89 (85.6%)	32 (84.2%)	
	>67	11 (10.6%)	6 (15.8%)	
CT FIB-test (55–87 s), mean		67.5 (45.2–86.5)	63.0 (47.8–73.5)	0.332 (0.740) **
CT FIB-test (>70 s), no. of patients (%)		46 (44.2%)	12 (31.6%)	1.357 (0.185) ***
A5 FIB-test (6–21 mm), mean ± SD		22.62 ± 6.43	27.45 ± 6.58	3.939 (˂ 0.0001) *
A5 FIB-test (>9 mm), no. of patients (%)		103 (99.0%)	38 (100.0%)	0.000 (1.000) ***
A10 FIB-test (7–23 mm), mean ± SD		17.48 ± 4.67	36.26 ± 4.46	21.472 (˂0.0001) *
A10 FIB-test (>23 mm), no. of patients (%)		13 (12.5%)	37 (97.4%)	84.189 (˂0.0001) ***
MCF FIB-test (9–27 mm), mean ± SD		24.62 ± 7.63	30.90 ± 7.91	4.294 (˂0.0001) *
MCF FIB-test (>25 mm), no. of patients (%)		48 (46.2%)	28 (73.7%)	7.409 (0.004) ***

Abbreviation: * t-t test ** Z-Z test *** χ2-chi-square test.

**Table 2 medicina-59-01202-t002:** Optimal cutoff values for a number of inflammation and coagulation parameters with the highest sensitivity and specificity in discriminating patients with increased mortality.

	AUC (95% CI for AUC)	*p*	Cutoff	Sensitivity (%)	Specificity (%)
Age	0.637 (0.536–737)	0.013	68	71.1	64.4
Fibrinogen (g/L)	0.702 (0.591–0813)	0.000	9.14	71.1	74.0
Anti-Xa	0.487 (0.378–0.596)	0.816	0.4	55.3	47.1
INR	0.790 (0.696–0.884)	0.000	1.38	76.3	75.0
D-Dimer (ng/mL)	0.961 (0.930–0.991)	0.000	860	86.8	95.2
aPTT (s)	0.474 (0.373–0.575)	0.637	31.2	73.7	37.5
ADP (AU/min)	0.878 (0.816–0.940)	0.000	591	63.2	95.2
ASPI (AU/min)	0.848 (0.772–0.924)	0.000	728	65.8	97.1
TRAP (AU/min)	0.955 (0.923–0.987)	0.000	1180	76.3	99.0
PSEP (ng/mL)	0.779 (0.701–0.856)	0.000	335	89.5	74.0
CT EX-test (s)	0.364 (0.261–0.468)	0.013	37	100.0	1.0
A5 EX-test (mm)	0.644 (0.544–0.744)	0.009	57	50.0	77.9
A10 EX-test (mm)	0.674 (0.574–0.774)	0.002	64	50.0	82.7
MCF EX-test (mm)	0.632 (0.533–0.730)	0.016	59	97.4	27.9
CT FIB-test (s)	0.482 (0.381–0.583)	0.740	43	97.4	22.1
A5 FIB-test (mm)	0.704 (0.597–0.810)	0.000	28	68.4	78.8
A10 FIB-test (mm)	0.989 (0.968–1.000)	0.000	30	97.4	100.0
MCF FIB-test (mm)	0.703 (0.607–0.798)	0.000	36	42.1	94.2

**Table 3 medicina-59-01202-t003:** Correlation between various coagulation and inflammation parameters.

	FI	Anti-Xa	INR	DD	aPTT	ADP	ASPI	TRAP	PSEP	CT EX-Test	A5 EX-Test	A10 EX-Test	MCF EX-Test	CFT EX-Test	CT FIB-Test	A5 FIB-Test	A10 FIB-Test	MCF FIB-Test
Age	−0.058 0.497	0.226 0.007	0.284 0.001	0.176 0.036	0.274 0.001	−0.010 0.910	−0.104 0.219	0.092 0.276	0.114 0.175	0.273 0.001	0.066 0.438	0.098 0.244	0.117 0.165	−0.029 0.731	0.390 0.000	0.208 0.013	0.338 0.000	0.152 0.071
Fibrinogen (g/L)		0.046 0.584	0.011 0.899	0.261 0.002	0.296 0.000	0.078 0.354	0.156 0.064	0.181 0.031	0.073 0.391	−0.274 0.001	0.432 0.000	0.405 0.000	0.396 0.000	−0.177 0.035	−0.390 0.000	0.680 0.000	0.110 0.195	0.613 0.000
Anti-Xa			0.042 0.622	−0.026 0.757	0.290 0.000	0.038 0.653	−0.043 0.612	−0.006 0.940	−0.033 0.696	0.062 0.467	0.006 0.942	0.011 0.897	−0.018 0.834	0.042 0.621	0.076 0.366	0.143 0.089	0.029 0.731	0.089 0.290
INR				0.351 0.000	0.093 0.272	0.249 0.003	0.192 0.022	0.218 0.009	0.144 0.088	−0.158 0.061	0.184 0.029	0.145 0.086	0.090 0.288	−0.167 0.047	−0.107 0.207	0.305 0.000	0.500 0.000	0.334 0.000
D-Dimer					0.146 0.083	0.487 0.000	0.513 0.000	0.555 0.000	0.196 0.019	0.037 0.660	0.341 0.000	0.362 0.000	0.319 0.000	−0.150 0.075	−0.007 0.933	0.357 0.000	0.572 0.000	0.371 0.000
aPTT (s)						−0.031 0.711	0.087 0.305	−0.012 0.883	−0.090 0.288	0.283 0.001	0.157 0.062	0.160 0.057	0.227 0.006	−0.090 0.286	0.069 0.411	0.421 0.000	−0.048 0.572	0.461 0.000
ADP (AU/min)							0.838 0.000	0.768 0.000	0.175 0.038	−0.135 0.106–9	0.392 0.000	0.444 0.000	0.346 0.000	−0.291 0.000	−0.178 0.034	0.147 0.081	0.532 0.000	0.312 0.000
ASPI (AU/min)								0.794 0.000	0.159 0.058	−0.266 0.001	0.490 0.000	0.574 0.000	0.490 0.000	−0.359 0.000	−0.362 0.000	0.306 0.000	0.424 0.000	0.414 0.000
TRAP (AU/min)									0.184 0.029	−0.247 0.003	0.426 0.000	0.422 0.000	0.440 0.000	−0.286 0.001	0.206 0.014	0.259 0.002	0.627 0.000	0.313 0.000
PSEP (pg/mL)										−0.080 0.344	−0.022 0.793	−0.001 0.992	−0.021 0.801	0.024 0.778	0.069 0.415	0.042 0.620	0.351 0.000	0.034 0.690
CT EX-test (s)											−0.399 0.000	−0.355 0.000	−0.277 0.001	0.315 0.000	0.722 0.000	−0.351 0.000	−0.127 0.131	−0.319 0.000
A5 EX-test (mm)												0.907 0.000	0.933 0.000	−0.770 0.000	−0.562 0.000	0.663 0.000	0.265 0.001	0.703 0.000
A10 EX-test (mm)													0.883 0.000	−0.728 0.000	−0.558 0.000	0.624 0.000	0.269 0.000	0.691 0.000
MCF EX-test (mm)														−0.772 0.000	−0.463 0.000	0.633 0.000	0.279 0.001	0.665 0.000
CT FIB-test (s)																−0.429 0.000	0.089 0.294	−0.421 0.142
A5 FIB-test (mm)																	0.294 0.000	0.917 0.000
A10 FIB-test (mm)																		−0.329 0.000

**Table 4 medicina-59-01202-t004:** Binary logistic regression of mortality predictors.

	Univariate		Multivariate	
	OR (95% CI for OR)	*p*	OR (95% CI for OR)	*p*
Age (≥68)	4.445 (1.981–9.970)	0.000		
Fibrinogen (g/L)	1.392 (1.093–1.773)	0.007		
Fibrinogen (>4 g/L)	0.083 (0.009–0.764)	0.028		
Fibrinogen (≥9.14 g/L)	7.000 (3.062–16.003)	0.000		
INR	486.244 (41.870–5646.800)	0.000		
INR (≥1.38)	9.667 (4.051–23.065)	0.000		
D-Dimer (ng/mL)	1.006 (1.004–1.008)	0.000		
D-Dimer (>1000 ng/mL)	125.182 (26.168–598.831)	0.000		
D-Dimer (≥860 ng/mL)	130.680 (35.590–479.835)	0.000	23.735 (2.824–199.461)	0.004
ADP test (AU/min)	1.011 (1.007–1.016)	0.000		
ADP test (<normal range>)	17.007 (5.085–56.873)	0.000		
ADP test (>590 AU/min)	33.943 (11.139–103.433)	0.000		
ASPI test (AU/min)	1.011 (1.007–1.016)	0.000		
ASPI test (<normal range>)	157.933 (19.847–1256.730)	0.000		
ASPI (≥728 AU/min)	64.744 (17.131–244.686)	0.000		
TRAP test (AU/min)	1.006 (1.004–1.009)	0.000		
TRAP test (<normal range>)	23.421 (8.669–63.278)	0.000	21.983 (2.365–204.311)	0.001
TRAP test (≥1180 AU/min)	331.889 (40.372–2728.392)	0.000		
PSEP (pg/mL)	1.003 (1.001–1.004)	0.000		
PSEP (>337 pg/mL)	18.822 (6.668–53.131)	0.000		
PSEP (≥335 pg/mL)	24.241 (7.870–74.663)	0.000		
CT EX-test (s)	0.970 (0.947–0.994)	0.014		
CT EX-test (<normal range>)	0.376 (0.186–0.763)	0.007		
A5 EX-test (mm)	1.100 (1.032–1.173)	0.003		
A5 EX-test (≥57 mm)	3.522 (1.604–7.734)	0.002		
A10 EX-test (mm)	1.121 (1.035–1.215)	0.005		
A10 EX-test (<normal range>)	3.985 (1.300–12.211)	0.016		
A10 EX-test (≥64 mm)	4.778 (2.117–10.782)	0.000		
MCF EX-test (mm)	1.107 (1.026–1.195)	0.009		
MCF EX-test (≥59 mm)	14.307 (1.875–109.149)	0.010		
A5 FIB-test (mm)	1.128 (1.056–1.204)	0.000		
A5 FIB-test (≥28 mm)	8.076 (3.521–18.525)	0.000		
A10 FIB-test (mm)	1.895 (1.410–2.546)	0.000		
A10 FIB-test (>23 mm)	259.000 (32.697–2051.584)	0.000	289.509 (3.438–24378.575)	0.012
MCF FIB-test (mm)	1.111 (1.053–1.172)	0.000		
MCF FIB-test (>25 mm)	3.267 (1.441–7.406)	0.005		
MCF FIB-test (≥36 mm)	11.879 (4.173–33.810)	0.000		

## Data Availability

Data are unavailable due to patient privacy, whose regulations are present in the hospital protocols of the Clinical Center Nis, Serbia.

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
