# Peer review of "Monitoring the Coagulation Profile of COVID-19 Patients Using Standard and ClotPro® Hemostasis Tests"

_medicina, 2023, doi:10.3390/medicina59071202_

Round 1

Reviewer 1 Report

In the submitted paper the authors describe a study performed on COVID-19 ARDS patients and identified, by appropriate statistics, several hemostatic parameters that were associated with higher mortality.

However, there are some important issues that need to be addressed:

1.     Abstract

The aim of the study should be included in the abstract.

The paragraph “Among all parameters assessed, mortality was associated with higher age (p<0.05), higher factor I (p<0.05), INR (p<0.001), D-Dimer (p<0.001), ADP (p<0.001), ASPI (p<0.001), TRAP (p<0.001), PSEP (p<0.001), A5 extest , (p<0.01), A10 extest (p<0.01), A5 fib (p<0.001), A10fib (p<0.001) and MCF fib (p<0.001) values, but lower CT extest values (p<0.05). Mortality was associated with extreme values of D-Dimer above 1000 (p<0.001), ADP above 590 (p<0.001), ASPI above 800 (p<0.001), TRAP above 1500 (p<0.001) and PSEP above 1000 (p<0.05)” needs to be reformulated in order to highlight the most important results. A suggestion is: “Among all parameters assessed, older age, increased levels of FI, INR, D-dimer, and PSEP, higher results of platelet aggregation tests (ADP, ASPI, TRAP) and some assays of the viscoelastic test (A5 EX-test, A10 EX-test, A5 FIB, A10 FIB, and MCF FIB), and lower values of viscoelastic CT EX-test, were significantly correlated with mortality. In multivariate analysis, D-dimer levels above 860 ng/ml, higher TRAP test value bins, and values above the normal reference range of A10 FIB-test, were found to be independent predictors of mortality”.

 2.     Introduction

Some information about the types of hemostasis tests that are used in the study should be added. Also, the aim of the study is missing. 

3.     Results: Please, provide more explanations for Table 1. For example, Factor I (g/L), values are expressed as mean ± SD; Factor I (>4 g/L), no. of patients (%), etc. Replace p=0.000  with p<0.0001.

4.     Discussion: This section needs to include discussion about similar coagulation studies performed on Covid-19 patients with ARDS. E.g. Heubner, L., Greiner, M., Vicent, O. et al. Predictive ability of viscoelastic testing using ClotPro® for short-term outcome in patients with severe Covid-19 ARDS with or without ECMO therapy: a retrospective study. Thrombosis J 20, 48 (2022).

      References are missing for some paragraphs (e.g. TRAP test).   

Author Response

Dear Reviewer 1,

First of all, I want to thank you for reviewing our article. Before I upload the final version of the article, I will give a detailed answer to your review. It goes without saying that I have adopted all your corrections and recommendations.
1. aim of the study is included in the abstract
2. I have reformulated paragraph in abstract as you suggested
3. In introduction i added the types of hemostasis tests that are used in the study and the aim of the study.
4. In results i provided more explanations for Table 1. and replaced p=0.000 with p<0.0001.
5. In discussion i included similar coagulation studies performed on Covid-19 patients with ARDS.
6. I added missing references for some paragraphs.
Thank you again.
Kind Regards
Milan Lazarevic, MD, Assist Prof

Reviewer 2 Report

Better description of patient population is needed.

Better explanation of assays used is needed.

Specific comments are related to each manuscript section.

Title

You must be more specific which VEA was used

Consider changing "viscoelastic point of care tests"  to the Clot -Pro

Abstract Results

Factor 1 is an outdated term for fibrinogen, consider using fibrinogen instead

ADP , ASPI, TRAP parameters method needs to be ID : aggregation with….

Similar for the Clot Pro parameters- please be consistent in the terminology

Units must be added to the parameter values

Key words

Consider adding Clot Pro

Material and methods

Please characterize patients population better with information relevant tor the Covid risk factors

Length of Covid symptoms prior to the ICU admission

- how many females, male

- comorbidities such as DM , immunocompromised , renal disease

- Assays used  in the study include anti Xa and plt agrregometry . Please explain  if  patients were on anticoagulants / antiplatelet agents ?

- Cause of death?

- how many patient developed DVT, PE ?

- average LOS in ICU

For assay descriptions that reader is not that familiar you need to provide brief description of the assays and significance of the parameters that are used for the study : Clot Pro and Plt aggregometry assays

If you adding units to the assays , be consistent : CRP line 69 ? line 75 parameters

Companies must be ID by the company name , city , state, country _ line 78

Factor I term for the fibrinogen is rarely used, consider using fibrinogen

Similarly if they received antiplatelet therapy , it has to be clarified

  Table 1

  Please specify RR for fibrinogen, PT, aPTT and all other lab values

Results

Consider  describing results better .

For example instead " mortality was associated " … consider " higher mortality…"Line 109, line 116, line 126

? survival …better or worst …not clear…line 118

Discussion

Line 210 - Your statement "Cut-off value of D-dimer, which is roughly four times larger than  normal, reflects hypercoagulability over hypofibrinolysis in critical ill Covid-19 patients"  is questionable and you did  not assess hypofibrinolysis .

> DD indicates that coagulation cascade was activated, clot was formed and fibrinolytic system was activated  as well …so yes this supports that hyperfibrinolysis is not that likely , but it does not mean that it reflects hypercoagulability , because DD can also be high in a bleeding patient.

You need to elaborate more on the meaning of A10 fib test - line 224

Study limitations

As authors pointed out fibrinolysis aberration is key pf increased mortality in Covid.

Clot Pros has AP test and TPA test to detect fibrinolysis or its alteration, but they were not used in this study, which is one of the study limitations. Please elaborate more why tests were not used in  your study.

Better description of patient population is needed.

Better explanation of assays used is needed.

Specific comments are related to each manuscript section.

Title

You must be more specific which VEA was used

Consider changing "viscoelastic point of care tests"  to the Clot -Pro

Abstract Results

Factor 1 is an outdated term for fibrinogen, consider using fibrinogen instead

ADP , ASPI, TRAP parameters method needs to be ID : aggregation with….

Similar for the Clot Pro parameters- please be consistent in the terminology

Units must be added to the parameter values

Key words

Consider adding Clot Pro

Material and methods

Please characterize patients population better with information relevant tor the Covid risk factors

Length of Covid symptoms prior to the ICU admission

- how many females, male

- comorbidities such as DM , immunocompromised , renal disease

- Assays used  in the study include anti Xa and plt agrregometry . Please explain  if  patients were on anticoagulants / antiplatelet agents ?

- Cause of death?

- how many patient developed DVT, PE ?

- average LOS in ICU

For assay descriptions that reader is not that familiar you need to provide brief description of the assays and significance of the parameters that are used for the study : Clot Pro and Plt aggregometry assays

If you adding units to the assays , be consistent : CRP line 69 ? line 75 parameters

Companies must be ID by the company name , city , state, country _ line 78

Factor I term for the fibrinogen is rarely used, consider using fibrinogen

Similarly if they received antiplatelet therapy , it has to be clarified

  Table 1

  Please specify RR for fibrinogen, PT, aPTT and all other lab values

Results

Consider  describing results better .

For example instead " mortality was associated " … consider " higher mortality…"Line 109, line 116, line 126

? survival …better or worst …not clear…line 118

Discussion

Line 210 - Your statement "Cut-off value of D-dimer, which is roughly four times larger than  normal, reflects hypercoagulability over hypofibrinolysis in critical ill Covid-19 patients"  is questionable and you did  not assess hypofibrinolysis .

> DD indicates that coagulation cascade was activated, clot was formed and fibrinolytic system was activated  as well …so yes this supports that hyperfibrinolysis is not that likely , but it does not mean that it reflects hypercoagulability , because DD can also be high in a bleeding patient.

You need to elaborate more on the meaning of A10 fib test - line 224

Study limitations

As authors pointed out fibrinolysis aberration is key pf increased mortality in Covid.

Clot Pros has AP test and TPA test to detect fibrinolysis or its alteration, but they were not used in this study, which is one of the study limitations. Please elaborate more why tests were not used in  your study.

 Overall article needs proofreading by the English native speaker proficient in medical terminology , authors also need to be consistent in terminology ( Ex , COVID 19 , COVID-19) , and use correct terminology . There are many topographical errors, misspelled words, incorrect sentence composition, punctuation.

Author Response

Dear Reviewer 2,

First of all, I want to thank you for reviewing our article. Before I upload the final version of the article, I will give a detailed answer to your review. It goes without saying that I have adopted all your corrections and recommendations.
1. I have changed tittle as you suggested to ClotPro (instead of VET)

2. I replaced F I term with fibrinogen not only in abstract but in whole article

3. I added units to all parameters, and i have explained used methods of Multiplate and ClotPro

4. I added ClotPro in key words

5. I better described patients population better with information relevant tor the Covid risk factors and Length of Covid symptoms prior to the ICU admission.

6. Patients were on prophylactic dose of LMWH. Patients didn t  receive AT therapy, but we done aggregometry to estimate effect of platelets on hypercoagulability and potential risks on thrombosis.

7. Causes of deaths were pulmonary embolism and arterial thrombosis, i added number of deaths and average LOS in ICU

8. I have provided brief description of the assays and significance of the parameters:  Clot Pro and Multiplate assays

9. I added units to the assays , and ID for companies

10. I added reference ranges for lab values in table 1.

11. I described results better as you suggested

12. In discussion elaborated more on the meaning of A10 fib test, and also i have excluded questionable statement about d-dimer

13. ClotPro tPA test was not used in  your study because it is not registered in our country

14. I send article for proofreading by the English native speaker proficient in medical, suggested by Medicina Journal

Thank you again.
Kind Regards
Milan Lazarevic, MD, Assist Prof

Reviewer 3 Report

I have carefully reviewed the article entitled "The coagulation profile monitoring of COVID-19 patients with standard and viscoelastic point of care hemostasis tests" and would like to provide some comments for revision.

1. Introduction: It is essential for the authors to provide a clear rationale for conducting this study and state the objectives. Additionally, a brief description of the viscoelastic test and the reasons for its utilization in this study should be included.

2. Methods: I would appreciate clarification on why the authors specifically included patients aged between 36-84 years and excluded younger or older individuals. Providing a justification for this age range will strengthen the methodology.

3. The article did not mention obtaining Research Ethics Board (REB) approval. It is crucial to confirm whether REB approval was obtained before the commencement of patient enrollment. 

4. If patients were intubated, it is important to explain how the authors obtained informed consent.

5. Please clarify whether the blood samples were collected on the first day of ICU admission or the first day of hospitalization. 

6. The authors did not mention the clinical variables that were collected as part of the data collection process. Including a list of the specific clinical variables.

7. Please provide a description of the primary outcomes of the study. 

8. Are there any exclusion criteria that were applied in this study?

9. I request the authors to provide a flow diagram illustrating the patient enrollment process. T

10. The authors should include baseline clinical characteristics of the patients. This should include information such as co-morbidities, current medication, COVID-19 severity, mode of mechanical ventilation, presence of COVID-pneumonia, presence of COVID-ARDS, renal impairment, liver impairment, use of anticoagulants (including dosage, whether prophylactic or therapeutic), use of aspirin, and recent history of VTE, arterial thrombosis, MI, stroke, etc.

11. Please provide a brief description of the treatment of COVID-19 employed in this study.

12. It would be helpful to specify when the assessment of mortality was conducted. 

13. The conclusion should be revised to reflect the main findings of this study. This paragraph is irrelevant and should be removed 

The key to success in the treatment of COVID-19 infection is timely and adequate 251
therapy and patient monitoring, which is impossible without early risk stratification and
mortality prediction.
252

none

Author Response

Dear Reviewer 3,

First of all, I want to thank you for reviewing our article. Before I upload the final version of the article, I will give a detailed answer to your review. It goes without saying that I have adopted all your corrections and recommendations.

  1. The rationale for studying coagulation profile monitoring in COVID-19 patients stems from observations of abnormal blood clotting and increased risk of thrombotic events in individuals with severe COVID-19.

The aim of the research is to: 1) examine the ability of routine parameters of hemostasis and the Clot Pro test to predict mortality; 2) examine the association of the examined parameters; 3) find the cut-off values of parameters that show predictive potential; 4) to test the predictive ability of combinations of obtained parameters.

2. Age less than 36 and more than 84 years was not an exclusion criterion, but it is the age distribution of patients who entered the study, who were ICU admitted in the period from 14th April 2020 to 25th May 2020.

3. REB approval was obtained before the commencement of patient enrollment, and it was the main condition among others for the article to be submitted to Medicina Journal. Number of REB protocol, date and also informed consent for patients from the study were sent to editors of Journal at the time of submission.

  1. Informed consent was obtained from each patient included in the study, before admission or during his admission to the ICU, certainly during the period of worsening of the patient's general condition but before the period of invasive mechanical ventilation.

5. The blood analyzes on the basis of which we obtained the data were taken during admission to the ICU.

6.-10. It will be clarified in the text

All patients admitted to the ICU had radiologically verified ARDS and required invasive mechanical ventilation. During the period of intensive treatment, they underwent continuous invasive monitoring of arterial and central venous pressure, monitoring of urinary output as well as depth of sedation (BIS). As we were guided by the principle of individualization of mechanical ventilation, we will try in several items to at least match the initial settings of mechanical ventilation. They were ventilated according to the principles of mechanical ventilation for ARDS in a pressure-controlled fashion: tidal volumes up to 6 ml/kg of ideal body weight, ventilation frequency determined according to carbon dioxide and pH value of arterial blood, with PEEP determined after the initial recruitment maneuver, and based on the improvement of compliance, arterial oxygenation or alveolar dead space. None of the patients included in the study had pre-existing thrombotic or bleeding disorders, pre-existing acquired coagulopathies, chemotherapy or active malignancy, renal or liver disease or pregnancy, and the listed comorbidities were exclusion criteria. Patients developed some of the above-mentioned conditions (AKI - acute kidney injury) as a complication of a critical illness. All patients received a prophylactic dose of low-molecular-weight heparin (LMWH) according to their body mass. Clinical variables monitored during study were: venous thromboembolism, pulmonary thromboembolism, and arterial thrombosis.

11. Patients were sedated by continuous administration of propofol or dexmedetomidine with monitoring of the depth of sedation by bispectral index (BIS). Continuous administration of a non-depolarizing muscle relaxant was not routinely applied. All patients were ventilated in semi-recumbent position using a closed suctioning system. In order to prevent pressure ulcers, all patients were turned every two hours. All patients received a daily dose of pantoprazole of 40 to 80 milligrams intravenously. Enteral nutrition was applied within 48 hours of ICU admission, except in the presence of clear contraindications for it. All patients had some type of corticosteroid therapy (dexamethasone, methylprednisolone or hydrocortisone, which was most often given in septic shock until hemodynamic stabilization). Exposing patients to empiric antimicrobial therapy was for shortest time until the arrival of sampled cultures and conditions for de-escalation.

12. We assessed intrahospital mortality and correlated hemostasis parameters with it.

  1. Done.

Kind regards

Milan Lazarevic, MD, Assist Prof

Reviewer 4 Report

I would like to congratulate the authors on doing research on this very interesting topic. I do think that results are novel and there might be a clinical application of the same. However I do have some major concerns

1. In general, the manuscript would greatly benefit from language revisions. There multiple issues with words choice, sentence structure etc. Just to list one example ( there are too many to list them all). Line 47 " .. and procalcitonin, and also fibrinogen"- really does not read well. So the entire paper should be proofread by either professional editor or native speaker skilled in academic writing.

2. Introduction must list common condition seen in clinical practice that are associated with hypercoagulability- for example, cancer, autoimmune diseases, celiac disease and inflammatory bowel diseases. Please list these conditions and cite appropriate papers in this section.

3. Methods- why did you include only patients on mechanical ventilation?

4. In methodology- exclusion criteria have not been mentioned. If your cohort had a lot of patients with celiac disease, cancer, IBD or lupus for example this would be a significant co-founder and for that reason this might be a really big problem of your analysis

5. Results- ADP, ASPI abbreviations are being used without previously being explained. This must be corrected

6. Patients with BMI ( and sometimes NAFLD) have protracted low grade inflammation that can results in elevated inflammatory markers even in the absence of COVID . Have you taken into account patients' BMI? 

7. Finally discussion focuses of interpretation of the results, when in fact more should be discussed about pathophysiology and compare these findings with other recent studies on this topic

Significant improvement is required 

Author Response

Dear Reviewer 4,

First of all, I want to thank you for reviewing our article. Before I upload the final version of the article, I will give a detailed answer to your review. It goes without saying that I have adopted all your corrections and recommendations.

  1. Article is now edited  by professional editor skilled in academic writing, suggested by Medicina Journal, and i will upload that version.
  2. In introduction i have listed conditions which are associated with hypercoagulability (15,16, 17, 18)
  3. Patients on mechanical ventilation (MV) were included because acute respiratory distress syndrome (ARDS) patients on MV are at high risk for DVT because they are susceptible to general risk factors for VTE and also to those specific to the critically ill, such as advanced age, sedation, immobilization, insertion of a central venous catheter combined with a severe inflammatory response and hypercoagulable states.
  4.  I have mentioned exclusion criteria as you suggested
  5.  I have explained ADP, ASPI, TRAP, CT, A10, A5, MCF abbreviations 
  6.  We took into account patients' BMI, but we didnt find significant correlation between BMI and inflammation in our study group. We didnt analysed patients with NAFLD
  7. Discussion has been rewritten                                                               Thank you again.
    Kind Regards
    Milan Lazarevic, MD, Assist Prof

Round 2

Reviewer 1 Report

The authors have complied with the requests formulated by the reviewers. There are still some minor issues:

Please, mark Table 1 in the main text when Results are presented

Please, remove from the main text the phrases: "The average fibrinogen concentration; the aPTT, and PT values; the average age in years; the ADP, ASPI, and TRAP test values; and also the CT- exEX-test results are expressed as mean ± SD. The number of patients, and the concentrations of fi-  brinogen and D-dimer values are expressed as percentage (%). All other results which are not in the reference range are also expressed in percentage (%)" that don't provide clear information.

Instead, provide a proper description for each line (parameter) of the Table 1:

   - Age (years), mean ± SD

   - Fibrinogen levels (g/L), mean ± SD

   - Fibrinogen > 4 g/L, no. of patients (%)

   - Anti-Xa values, mean ± SD

   - INR values, mean ± SD

   - INR >1, no. of patients (%)

   - D-dimer levels, mean ± SD

   - D-dimer > 230 ng/mL, no. of patients (%)

   - D-dimer > 1000 ng/mL, no. of patients (%)

   - Replace "No mortality" with "Survival", etc...

Explain t, Z, χ2 in Table 1

Replace p=0.000 with p<0.0001

Author Response

Dear Reviewer 1, Here is my response to revision: -I included proper explanations in table 1. for each parameter instead of the text which didn't provide clear explanations  -I replaced no mortality with survival  -I added abbreviations for statistical tests -I replaced p=0.000 with p<0.0001 Thank you again for the revisions and your help

Reviewer 2 Report

No additional comments. Authors addressed all my suggestions. Thank You. 

Author Response

Thank you

Reviewer 3 Report

I am satisfied with authors' response. 

Author Response

Thank you

Reviewer 4 Report

I would like to thank the authors on detailed revision of the paper. I do not have any further comments. 

Author Response

Thank you